# Towards Informative Open-ended Text Generation with Dynamic Knowledge Triples

**Zixuan Ren[1,2], Yang Zhao[1,2], Chengqing Zong[1,2]***

[1] State Key Laboratory of Multimodal Artificial Intelligence Systems,
Institute of Automation, CAS, Beijing, China
[2] School of Artificial Intelligence, University of Chinese Academy of Sciences, Beijing, China
{renzixuan2021@, zhaoyang2015@, cqzong@nlpr.}ia.ac.cn

## Abstract

Pretrained language models (PLMs), especially large language models (LLMs) demonstrate impressive capabilities in open-ended text generation. While our statistical results show that LLMs often suffer from *over-concentrated information*, where the generated texts overly focus on the given prompt and fail to provide sufficient background and detailed information as humans do. To address this issue, we propose a dynamic knowledge-guided informative open-ended text generation approach, that utilizes a knowledge graph to help the model generate more contextually related entities and detailed facts. Specifically, we first employ a local knowledge filter to extract relevant knowledge from the comprehensive knowledge graph for a given topic sentence. Then we introduce a dynamic knowledge selector to predict the entity to be mentioned in the subsequent sentence. Finally, we utilize a knowledge-enhanced text generator to produce a more informative output. To evaluate the effectiveness of our approach, we evaluate the proposed approach in two scenarios: fine-tuning for small PLMs and prompt tuning for LLMs. Experimental results show that our approach could generate more informative texts than baselines.

## 1 Introduction

Open-ended text generation is a complex undertaking, encompassing the need for fluent sentences, a natural and coherent flow, as well as informative and non-trivial content (Yao et al., 2019). In recent years, much of the research has concentrated on addressing grammatical correctness concerns such as inter-sentence coherence and ensuring relevance between the input prompt and the generated story content (Yao et al., 2019; Guan et al., 2020, 2021; Tang et al., 2022). However, thanks to the advancements in large language models (LLMs) like GPT-3 (Brown et al., 2020) and

---
*Corresponding author.

Figure 1: Entity flow in LLM generated texts and human written texts. The entities not mentioned in input are highlighted with red and underline. The full texts are shown in Appendix A

ChatGPT (OpenAI, 2022), many of these aforementioned challenges have been substantially mitigated(Clark et al., 2021; Xie et al., 2023).

PLMs, particularly LLMs, have made significant advancements in open-ended text generation. However, our analysis reveals that current LLMs still face the challenge of over-concentrated information, wherein they prioritize events and entities mentioned in the prompt and fail to provide relevant entities and sufficient information as humans would. We define over-concentrated information as follows:

***Over-concentrated Information:***

*Texts that predominantly concentrate on the events and entities mentioned in the prompt often face challenges in encompassing relevant entities' comprehensive information.*

This discrepancy is evident in Figure 1, where we compare texts generated by human writers

| Name | Informativeness avg rank | # entity |
|---|---|---|
| human | 1.60 | 61.75 |
| GPT-3.5-davinci | 2.75 | 16.60 |
| GPT-3.5-turbo | 3.25 | 14.05 |
| LLAMA-13B | 4.00 | 40.95 |
| GPT-3.5-curie | 4.40 | 10.80 |
| GPT-3-davinci | 4.60 | 39.45 |

Table 1: The human evaluation results on informativeness comparison. Informativeness avg rank means the average rank of the model. # entity means the average number of entities in the text.

with those generated by powerful LLMs such as `GPT-3.5-turbo` and `Text-Davinci-003` using the headline "Elon Musk sets low expectations before first SpaceX launch of Starship, most powerful rocket ever built". Human-written text not only includes information directly related to the headline but also provides additional contextual details, such as the collaboration between NASA, FAA, and SpaceX, as well as specific information about the rocket and the launch location. On the other hand, texts generated by `GPT-3.5-turbo` and `Text-Davinci-003` tend to be focused solely on the provided information and lack the depth of additional relevant details beyond the headline.

In addition to analyzing specific cases, we also conducted human evaluation to compare the informativeness of texts written by humans with those generated by LLMs.[1] We select a diverse range of PLMs and ask annotators to rank the texts based on the informativeness. The results are listed in Table 1. The results demonstrate that human-written texts outperform all the tested large language models (LLMs) in terms of informativeness.

To address the issue of *over-concentrated information*, we propose a dynamic knowledge-guided informative open-ended text generation approach, **InfoGen**, in which a knowledge graph is utilized to help the model generate more related entities and detailed facts. InfoGen consists of several key components. First, we employ a local knowledge filter to extract relevant information from a comprehensive knowledge graph. Second, we introduce a dynamic knowledge selector, which predicts the entity to be mentioned in the subsequent sentence. Finally, we utilize a knowledge-enhanced text generator that leverages the filtered local knowledge and selected entities to produce a more informative

---

[1]Specific details are listed in Appendix B

and coherent output.

Furthermore, we construct a Chinese news generation dataset and annotate it, along with two existing English datasets, using named entity recognition and entity-linking techniques. To evaluate the effectiveness of InfoGen, we conduct experiments in two scenarios: fine-tuning for small PLMs and prompt learning for LLMs. The experimental results demonstrate that InfoGen consistently outperforms baseline models in terms of text informativeness. Our contributions are summarized as the following points:

- We raise the informativeness problem of open-ended text generation and propose a generation model that effectively utilizes knowledge for generating informative open-ended texts.

- We contribute a Chinese news generation dataset *ChinaNews* that includes entity annotations, along with the annotation of two existing English datasets.

## 2 Methodology

### 2.1 Framework

In this section, we introduce the framework of InfoGen. As is shown in Figure 2, InfoGen primarily consists of three modules.

- Local knowledge filter, a model to filter relevant knowledge from the knowledge graph based on the input information.

- Dynamic entity selector, a model to filter the entities that need to be mentioned in the next step based on the headline and generated text.

- Knowledge fused text generator, a model to generate text iteratively based on the headline, knowledge graph, previously generated text, and the entities that need to be mentioned.

**Local Knowledge Filter with Contrastive Learning** We employ a bi-encoder architecture consisting of two encoders, EncX and EncT, to separately encode the headline and knowledge triples. The encoding representation is obtained from the hidden states of the [CLS] token. Formally, the representation of the input headline $x$ and triple $t$ can be represented as:

$$H_X = \text{EncX}_{[\text{CLS}]}(X), \ H_t = \text{EncT}_{[\text{CLS}]}(t) \quad (1)$$

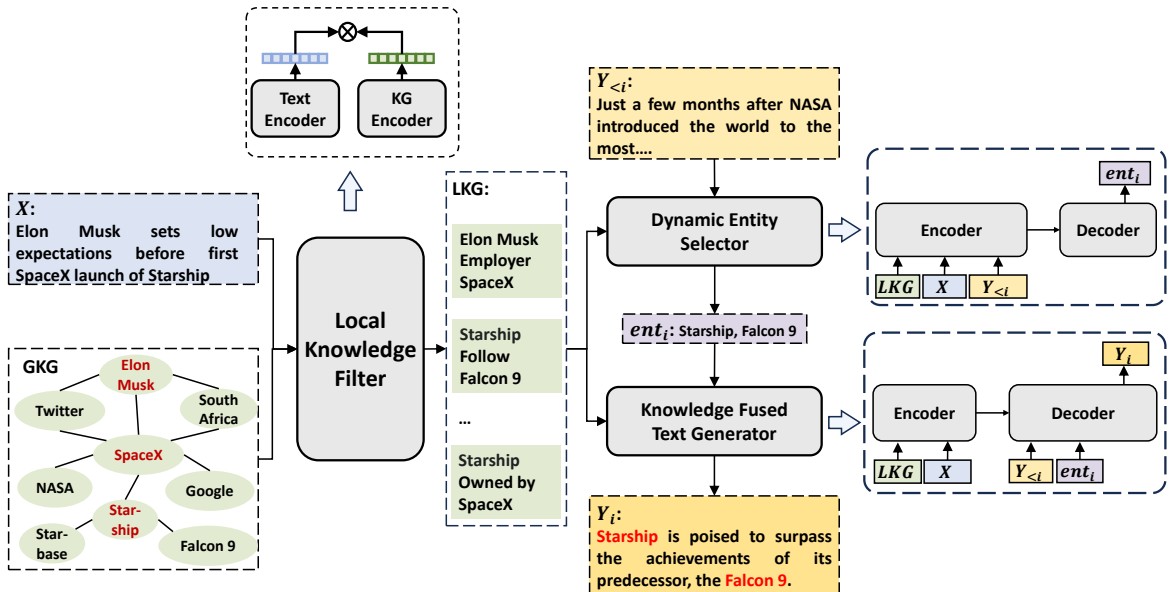

Figure 2: As InfoGen is iterative generative model, the picture describes the procedure of generating the i-th step. In this picture, $X$ denotes input, $Y_{<i}$ denotes generated texts and $Y_i$ denotes the target text. **GKG** denotes global knowledge graph and **LKG** denotes local knowledge filtered by Local Knowledge Filter. In GKG, entities mentioned in headline are highlighted. $ent_i$ denotes entities selected from local knowledge graph. We use arrow to show the specific structure of the models.

To measure the similarity between the headline and triple, we use the dot product as the similarity function:

$$\text{sim}(X, t) = H_X^T H_t \quad (2)$$

To filter the relevant knowledge triples, we employ contrastive learning. Given a headline, we consider the knowledge triples contained in the news article as positive knowledge samples, while other triples that include entities present in the headline or the news article are treated as negative samples. Let the positive knowledge triples be $P = \{p_1, \ldots, p_k\}$, and the negative knowledge triples be $N = \{n_1, \ldots, n_m\}$. Inspired by Karpukhin et al. (2020), we define our contrastive learning loss as follows:

$$\mathcal{L}(X, P, N) =$$
$$-\frac{1}{k} \sum_{i=1}^{k} \log \frac{\text{sim}(X, p_i)}{\text{sim}(X, p_i) + \sum_{j=1}^{m} \text{sim}(X, n_j)} \quad (3)$$

**Dynamic Entity Selector** The dynamic triple selector is designed to predict the entities that should be mentioned in the next step, given the headline and global knowledge. Inspired by generative named entity recognition approaches (Fries et al., 2017; Yan et al., 2020), we employ a generative approach for entity prediction. The auto-regressive

procedure is as follows:

$$P(e_{i_t} | e_{i_{<t}}) = \text{softmax}(\boldsymbol{W} H_{\text{Dec}} + \boldsymbol{b}) \quad (4)$$

Where $H_{\text{Dec}} = \text{Dec}(H_{<t}, Enc(X, Y_{<i}, T))$ is the hidden states from decoder. $e_{i_t}$ and $e_{i_{<t}}$ denote the t-th token and all previous tokens in the entity for the i-th sentence. $X$ and $Y_{<i}$ denote the input and generated texts. $H_{<t}$ represents the hidden states of previous tokens. Dec and Enc refer to the decoder and encoder of the model. $\boldsymbol{W}$ and $\boldsymbol{b}$ are learnable parameters.

To leverage the global knowledge, we adopt a modified encoder-decoder architecture called Fusion-in-Decoder (Izacard and Grave, 2021). This architecture encodes the headline and each triple separately and then concatenates the encoded representations for the decoder. Formally, $T = \{t_1, t_2, \ldots, t_k\}$ denotes the local knowledge triples. The Fusion-in-Decoder can be defined as:

$$\text{Enc}(X, Y_{<i}, T) = \text{concat}(\{H_{t_i}\}_1^k, H_{XY})$$
$$H_{t_i} = \text{Enc}(t_i) \quad (5)$$
$$H_{XY} = \text{Enc}(X, Y_{<i})$$

During the inference stage, to ensure the generation of meaningful and accurate entities, we construct a trie-tree that contains the entities present in the global knowledge. This trie-tree serves as

a constraint on the generation process, helping to avoid the generation of nonsensical or incorrect entities.

**Knowledge Fused Text Generator.** Building upon previous plan-based story-telling approaches (Yao et al., 2019; Hu et al., 2022), we adopt a dynamic approach that combines entity prediction and text generation. Specifically, $Y_{<i}$ and $e_{<i}$ denote the generated texts and their plan respectively and $e_i$ denotes the plan for the target text. The context $C_i$ is represent as" $e_1$$Y_1$$\cdots$ $e_{i-1}$$Y_{i-1}$$e_i$", in which "" represents a special separate token. The iterative procedure is defined as follows:

$$P(Y_{i_t}|C_i, X, T) = \text{softmax}(\mathbf{W}H_{\text{Dec}} + \mathbf{b}) \quad (6)$$

Here $H_{Dec} = \text{Dec}(C_i, \text{Enc}(X, T))$ is the decoder hidden states and the encoder is the same as in equation 5 .

## 2.2 Fine-tuning Small Language Model

Having trained the Local Knowledge Filter and Dynamic Entity Selector, we proceed to fine-tune small PLMs as the Knowledge Fused Text Generator. Considering the strong local modeling capabilities but potential lack of long-range coherence, we adopt a sentence-by-sentence approach for training and text generation. Let us suppose that the target texts $Y$ consist of $n$ sentences $Y_1, \cdots, Y_n$ with lengths $m_1, \cdots, m_n$, and the input comprises $X$ and selected knowledge $T$. Building upon the generation procedure defined in Equation 6 and the context definition of $C_i$, our training objective can be expressed as follows:

$$\arg\max_{\theta} \sum_{i=1}^{n} \sum_{t=1}^{m_i} \log P(Y_{i_t}|C_i, X, T, \theta) \quad (7)$$

Here, $\theta$ represents the parameters of the small PLMs, and we aim to maximize the log probability of generating each token $Y_{i_t}$ conditioned on the context $C_i$, input $X$, selected knowledge $T$, and the model's parameters.

## 2.3 Prompt Learning

We employ InfoGen to complement LLMs. Recognizing that LLMs themselves can be considered as vast knowledge bases (Petroni et al., 2019; Teubner et al., 2023) and it is hard to modify the architecture of LLMs, we integrate InfoGen as a text generator guided by the dynamic entity selector. LLMs are not finetuned to incorporate with plan and sentence level plan could make the LLMs pay too much attention to the entities. In this regard, we utilize the following prompt to drive the generation process:

*"Now you are a news writer. Please generate news article paragraph by paragraph based on the given headline. Before each paragraph generation, you may be given several entities, your generation needs to concern the entity. "*

Subsequently, we provide the headline along with the entities predicted by the dynamic entity selector. Following the generation process, we feed the generated texts back to the dynamic entity selector. If the dynamic entity selector produces a null output, we prompt the model to continue without any specific entity guidance.

The inference algorithm of §2.2 and §2.3 are demonstrated in algorithm 1.

---

**Algorithm 1:** Algorithm of inference stage of InfoGen

**Input:** Input $X$; Knowledge graph triples
$G = t_1, t_2, \cdots, t_n$; Knowledge number $k$
**Output:** Generated text $Y$

1   $Y \leftarrow$ ""
     // Filter local knowledge $LK$
2   $scores = []$
3   **foreach** $t$ *in* $G$ **do**
4      |   $scores.append(similarity(X, t))$
5   **end**
6   $s \leftarrow sorted(zip(G, scores))$
7   $LK \leftarrow s[:k]$
     // Iterative text generation
8   **while** *True* **do**
9      |   $e \leftarrow Entity\_selector(X, LK, Y)$
10     |   $text \leftarrow Text\_generator(X, e, LK, Y)$
11     |   **if** $text \neq null$ **then**
12     |    |   $Y.extend(text)$
13     |   **end**
14     |   **else**
15     |    |   break
16     |   **end**
17   **end**

---

## 3 Datasets Construction

We conduct evaluations of InfoGen on three news generation datasets: PENS (Ao et al., 2021), which comprises news articles collected from Microsoft News; CNN (Hermann et al., 2015), which includes news sourced from CNN websites; and ChinaNews, a dataset we construct, consisting of global news collected from Chinese official news agencies.

**Chinese Datasets Collection** To evaluate the effectiveness of InfoGen on Chinese text generation, we construct a Chinese news generation dataset. To

|  | PENS | CNN | ChinaNews |
|---|---|---|---|
| # Document | 113218 | 92465 | 236769 |
| # Input | 10.36 | 40.03 | 21.85 |
| # Output | 555.51 | 601.02 | 463.64 |
| # Entity | 39.99 | 36.02 | 21.62 |
| # Triple | 64.53 | 32.02 | 43.12 |

Table 2: Statistics of our dataset. # Document denotes the number of examples. # Input and # Output lines display the average number of tokens in input and output texts, respectively. # Entity and # Triple indicate the average number of entity and knowledge triple in the output text.

ensure the quality and reliability of the news data, we source articles from reputable Chinese official news agencies, including People's Daily, China Daily, and Xinhua News Agency. We focused on global news rather than local or entertainment news. The collection period for the dataset spanned from January 1, 2017, to August 1, 2022. After gathering the news articles, we performed a thorough preprocessing step, which involved filtering out hyperlinks, images, image captions, and any advertisements present in the corpus. Additionally, we removed brief news reports that contained less than 100 words, ensuring that the dataset consisted of substantive news articles suitable for our evaluation.

**Datasets Automatic Annotation**  To facilitate the utilization of knowledge graphs, we employ NLP automatic annotation tools to annotate the news datasets. For the English datasets PENS and CNN, we first filter noisy samples such as empty samples or samples containing too many advertisements and hyperlinks. Then we utilized Flair (Akbik et al., 2018) for entity recognition and BLINK (Ledell Wu and Zettlemoyer, 2020) to establish entity links with the Wikidata knowledge graph [2]. In the case of ChinaNews, since Flair and BLINK do not support Chinese, we employed SpaCy (Honnibal et al., 2020) for entity recognition and linked them to Wikidata using the Wikidata query API [3]. For relation extraction, we use distant supervision (Mintz et al., 2009) with Wikidata. The dataset statistics are provided in Table 2. For each dataset, we randomly choose 2000 samples as the validation set and 500 samples for the test.

---

[2] We use the Wikidata dumps archived on October 4, 2022
[3] https://query.wikidata.org/

| Metric | Spearman's $\rho$ | Kendall's $\tau$ | Pearson's $r$ |
|---|---|---|---|
| entity | 0.398* | 0.313* | 0.387* |
| Length | 0.487* | 0.396* | 0.479* |
| Event | 0.483* | 0.381* | 0.481* |
| KC-entity | **0.554**\*\* | **0.440**\*\* | **0.550**\*\* |

Table 3: Result of correlation tests between automated metric and human annotator ranking. * means the $p$ value is less than 1e-6 and ** means the $p$ value is less than 1e-10.

## 4 Experiments

### 4.1 Baselines

For comparison, we consider the following baselines: fine-tuned GPT-2 (Radford et al., 2019), fine-tuned BART (Lewis et al., 2020), and CK-GPT (Guan et al., 2020). GPT-2 and BART are pretrained language models commonly employed for natural text generation tasks. CK-GPT, on the other hand, is a variant of GPT-2 that incorporates commonsense knowledge by leveraging the ATOMIC(Sap et al., 2019) and Concept-Net(Liu and Singh, 2004) commonsense knowledge graphs.

### 4.2 Implementation Details

In the fine-tuning stage, we use Adam optimizer and set $\beta_1$=0.9, $\beta_2$=0.999, and $\epsilon$=1e-8. We train our model for 15 epochs and use the model with the best performance in the validation set for generation. During inference, we use nucleus sampling with $p$=0.9. We train our model using RTX-3090 GPUs on Ubuntu 18.04 and use NLP open-source library Transformers (Wolf et al., 2020). In our study, we employed a widely adopted pre-trained model based on the Transformer architecture as the foundational model(Zhao et al., 2023). Specifically, we use BERT(Devlin et al., 2019) as the base encoder for the local knowledge filter and BART-base(Lewis et al., 2020) as the base model for the dynamic entity selector. For fine-tuning PLMs, we choose Bart-base as a knowledge-fused text generator and for prompt learning, we choose GPT-3.5-turbo as the base model.

### 4.3 Evaluation Metrics

**Automatic evaluation metric**  (1) **Perplexity (PPL)**: smaller perplexity means model could generate more fluent texts. (2) **Distinct-3 (Dist-3)** (Li et al., 2016): the ratio of distinct 3-gram to generated 3 grams, which indicates the diversity of generated texts. (3) **ROUGE-L** (Lin, 2004): the recall of the longest common sub-sequence between gen-

| Models | CNN | | | | |
| --- | --- | --- | --- | --- | --- |
| | PPL↓ | Dist-3↑ | Rouge-L↑ | BERTScore↑ | KC-E↑ |
| GPT2 | 6.28 | 62.71 | 13.39 | 82.10 | 17.52 |
| BART | 6.11 | **63.62** | 13.64 | 81.78 | 14.65 |
| CK-GPT | 7.39 | 62.01 | 12.00 | 81.28 | 13.59 |
| INFOGEN | **5.49** | 63.59 | **14.53** | **82.91** | **22.16** |
| w/o DE | 5.81 | 63.22 | 14.19 | 82.73 | 20.08 |
| w/o LKG | 6.01 | 62.73 | 13.01 | 82.58 | 19.94 |
| | **PENS** | | | | |
| | PPL↓ | Dist-3↑ | Rouge-L↑ | BERTScore↑ | KC-E↑ |
| GPT2 | 6.83 | 61.94 | 10.05 | 81.83 | 5.58 |
| BART | 6.74 | 60.79 | 10.11 | 81.78 | 6.03 |
| CK-GPT | 7.54 | 58.61 | 9.67 | 79.09 | 5.52 |
| INFOGEN | **6.12** | **63.16** | **10.97** | **82.40** | **9.25** |
| w/o DE | 6.29 | 63.11 | 10.68 | 82.11 | 8.42 |
| w/o LKG | 6.57 | 62.17 | 10.49 | 81.94 | 8.04 |
| | **ChinaNews** | | | | |
| | PPL↓ | Dist-3↑ | Rouge-L↑ | BERTScore↑ | KC-E↑ |
| GPT2 | 5.36 | 53.64 | 17.08 | 79.17 | 13.49 |
| BART | 5.31 | 54.16 | 17.36 | 79.04 | 12.84 |
| INFOGEN | **4.52** | **55.84** | **18.25** | **80.01** | **17.52** |
| w/o DE | 4.83 | 55.05 | 18.02 | 79.72 | 16.22 |
| w/o LKG | 5.16 | 54.39 | 17.81 | 79.53 | 15.49 |

Table 4: Performance comparison of various models for open-ended news generation. Bold and underlined fonts indicate the best and second-best approaches, respectively. The symbol ↑ signifies that higher values are preferable, while ↓ signifies that lower values are preferable.**w/o DE** means ablating dynamic entity generator and **w/o LKG** means ablating local knowledge for news generation.

erated texts and reference texts. (4) **BERTScore** (Zhang et al.): the similarity of generated texts and reference texts measured by the cosine similarity of BERT embedding.

**Metric for Informativeness** Intuitively, the informativeness of generated texts can be evaluated using naive automated metrics including text length and the number of mentioned entities. However, due to the data memorization tendencies of LLMs (Carlini et al., 2021; Xie et al., 2023) and degeneration of PLMs(Holtzman et al., 2020; Dou et al., 2022a), the models can sometimes generate texts that have little or no relation to the input, which needs to be taken into consideration.Taking this into consideration, we propose the **Knowledge-constrained entity number (KC_entity)**, which only considers relevant entities. Formally, let $H = h_1, h_2, \cdots, h_m$ denote the entities in the headline, and $T = t_1, t_2, \cdots, t_n$ represent the entities in the generated text. Consider a knowledge graph $G$, where Rel(x, y) denotes the existence of a relationship between entities $x$ and $y$ in $G$. We define the knowledge-constrained entity set $A$ and

KC_entity with indicator function $\mathbb{I}$ as follows:

$$A := \{t \in T | \exists h \in H, \mathrm{Rel(t, h)}\}$$

$$\mathrm{KC\_entity} = \sum_{i=1}^{n} \mathbb{I}_{t_i \in A} |t_i| \tag{8}$$

We calculate the Pearson, Spearman, and Kendall correlation coefficients between human ranks and automated ranks.[4] We choose three basic metrics for informative measurement: number of words (Length) and number of entities (entity) and number of events (Event). The result is shown in Table 3.

**Human evaluation** To give a more comprehensive analysis, we conduct human evaluations that consider the following three aspects. (1) **Coherence (Coh.)**: evaluate whether the generated text contains off-prompt, self-contradiction, and redundant cases. (2) **Informativeness (Inf.)**: evaluate the informativeness of generated texts, including background information and details. (3) **Overall quality (Ovr.)**: take both coherence and informativeness into consideration.The annotators are asked to score 1 (worst) to 5 (best) for the three aspects.

---

[4]Here we reuse the human evaluation results in §1, details are listed in Appendix B

Each questionnaire contains 20 samples and is assigned to 3 annotators

# 5 Results and Analysis

## 5.1 Automatic Results of Fine-tuning Approach

We present the results of our automatic evaluation in Table 4. In the English datasets, KG-GPT performs worse than GPT2, which is consistent with previous findings (Wang et al., 2022). It has been noted that the incorporation of commonsense knowledge inference abilities can sometimes have a detrimental effect on the generation ability. In contrast, InfoGen demonstrates superior performance compared to the baselines across various metrics, including perplexity, Distinct-3, Rouge-L, and BERTScore. These results indicate that Info-Gen is capable of generating more coherent and fluent texts. The effectiveness of InfoGen can be attributed to the utilization of global knowledge and entities as a content plan for text generation. This strategy is beneficial in capturing long-range dependencies in previous studies (Xu et al., 2020; Guan et al., 2021). Furthermore, InfoGen outperforms the baselines in terms of the informativeness metric, as reflected by the KC-Entity scores. This improvement highlights the contribution of incorporating knowledge in generating more informative text.

## 5.2 Ablation Analysis

We conduct an ablation analysis of InfoGen, and the results are presented in Table 4. In the table, 'w/o DE' refers to the removal of the dynamic entity module, meaning that the text generator can only access local knowledge. 'w/o LKG' indicates the removal of local knowledge during text generation, while still retaining it for the local entity generator due to the trie-tree constrained generation.

The results of the ablation analysis confirm the effectiveness of both the local knowledge and dynamic entity components. When removing the local knowledge, we observe a decrease in performance, demonstrating the importance of incorporating local knowledge in generating informative texts. Additionally, there is a slight drop in performance when the dynamic entity module is removed. This suggests that global knowledge plays a more significant role in the overall performance of Info-Gen.

| | Coh. | Inf. | Ovr. |
|---|---|---|---|
| GPT-2 | 3.17 | 2.78 | 3.03 |
| BART | 3.15 | 2.82 | 3.02 |
| INFOGEN | **3.27** | **3.25** | **3.25** |

Table 5: Human evaluation of results of fine-tuning for small PLMs approach and baselines on CNN datasets. Coh., Inf. and Ovr. represents coherence, informativeness and overall quality respectively.

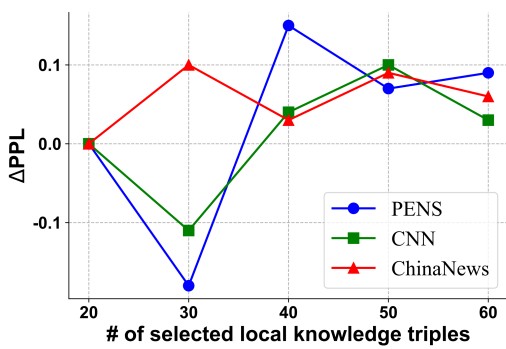

Figure 3: Impact of the number of selected knowledge

## 5.3 Human Evaluation of Fine-tuning Method

We ask human annotators to give scores to generated texts, from three aspects, including text coherence, text informativeness, and overall quality. The results of the human evaluation are presented in Table 5. The Fleiss' kappa is 0.41. Compare to baselines, InfoGen has a higher score in the three aspects.

## 5.4 Influence of Number of Local Triples

During the inference process, we employ a selection mechanism to choose the top-k triples based on the scores assigned by the local knowledge filter. To further investigate the impact of the selection parameter (k), we conduct an experiment using the validation dataset to observe the changes in perplexity ($\Delta PPL$) and the number of selected local knowledge. The results are depicted in Figure 3. In the PENS and CNN datasets, we observe that as the number of selected knowledge increases from 20 to 60, the perplexity initially decreases and then starts to rise. However, in the ChinaNews dataset, the perplexity consistently increases as the number of selected knowledge grows. Based on these findings, we determined the optimal values for k during the inference stage. Specifically, we chose k as 20 for ChinaNews and 30 for CNN and PENS.

| | Coh. | Inf. | Ovr. |
|---|---|---|---|
| GPT-3.5-turbo | 4.60 | 4.00 | 4.33 |
| GPT-3.5-turbo+Dynamic Entity | 4.60 | **4.75** | **4.72** |

Table 6: Human evaluation of texts generated by GPT-3.5-turbo and Dynamic Entity guided GPT-3.5-turbo.

## 5.5 Human Evaluation of Prompt Method

The results, displayed in Table 6 and achieving a Fleiss' kappa of 0.71, clearly illustrate that incorporating entity guidance does not sacrifice the coherence of the generated texts. Furthermore, it substantially enhances the informativeness of the generated content. For a detailed examination, please refer to the case study provided in Appendix D. By utilizing entity guidance, GPT-3.5-turbo successfully generates entities that are relevant to the topic, even if they are not explicitly mentioned in the headline. An example of this is the mention of the "Starbase Launch Site", which is one of the launch sites for the Starship spacecraft.

## 6 Related Work

**Knowledge Enhanced Open-ended Text Generation.** In recent years, there has been a growing emphasis on integrating knowledge into open-ended text generation, as evidenced by the introduction of various commonsense text generation datasets such as ROCStory (Liu et al., 2020). Researchers have directed their attention toward incorporating knowledge into the process. For instance, Guan et al. (2020) devised a method to convert knowledge triples into natural language using predefined templates and subsequently conducted post-training of GPT2 (Radford et al., 2019) on the resulting corpus. Another approach, inspired by the Plug-and-Play Language Model (Dathathri et al.), was pursued by Xu et al. (2020), who treated knowledge triples as control prefixes. Lin et al. (2022) generate future event of story by inferring commonsense explanation of generated context. However, these methods have primarily been tested on specially-designed datasets that focus on commonsense knowledge inference or understanding. Their approaches have not undergone extensive evaluation in the context of real-world open-ended generation tasks, such as opinion generation or news generation.

**Long Text Generation.** Previous research has highlighted the issue of off-prompt generation and incoherence in long-form text generation (Holtz-

man et al., 2020; Dou et al., 2022b). Existing approaches to solve these issues can be divided into two categories based on the generation mechanism. One approach involves generating text in a single pass while improving text coherence by leveraging the hierarchical structure of human-generated texts and modeling high-level semantic features. This includes considering key words (Rashkin et al., 2020), event sequences (Zhai et al., 2019), and sentence-level representations (Hu et al., 2022). Another category decomposes the text generation process into multiple stages. Some approaches generate keywords or an outline of the text first and then generate and refine the complete text (Yao et al., 2019; Hua and Wang, 2020; Tan et al., 2021; Yang et al., 2022). Other techniques employ latent vectors as continuous text plans (Shen et al., 2019; Ji and Huang, 2021; Tang et al., 2022). The recent success of LLMs has demonstrated that these issues can be effectively mitigated without the need for fine-tuning on specific datasets (Clark et al., 2021; Dou et al., 2022a; Xie et al., 2023). Despite this progress, it is still necessary to analyze and identify areas where LLMs may encounter limitations or fail to perform optimally.

**Text Informative Analysis.** In tasks involving rich input information such as translation, summarization, and data-to-text generation, informativeness assumes a significant role (Yuan et al., 2021). In summarization, informativeness serves as a metric to assess whether generated texts effectively capture the key information from the input context (Grusky et al., 2018; Zhu et al., 2020; Xiao et al., 2023). In data-to-text generation and machine translation, informativeness, also referred to as adequacy, gauges whether the generated texts convey the same information as the given input text (White, 1995; Lopez, 2008; Mehta et al., 2022). Currently, the evaluation of informativeness relies on human judgment, and the absence of automated metrics to quantify informativeness in creative text generation tasks remains a limitation.

## 7 Conclusion

In this paper, we address an important and unresolved issue in knowledge-intensive text generation, namely over-concentrated information. Through manual evaluations of news texts generated by both human writers and models, we discovered substantial disparities between the two, emphasizing the difficulty of generating informative

content. Drawing from these insights, we propose a dynamic knowledge-guided approach to open-ended text generation called InfoGen. Our method involves filtering relevant knowledge triples and dynamically selecting entities as content plans. We conduct experiments in two scenarios: fine-tuning small PLMs and prompt learning for LLMs. The results demonstrate that our approach effectively enhances the informativeness of the generated texts, thereby addressing the limitations observed in previous methods.

## Limitations

While our study focuses on knowledge-intensive open-ended text generation specifically in the context of news generation, it is important to acknowledge the limitations of our research. Firstly, knowledge-intensive open-ended text generation encompasses a wide range of application areas beyond news generation alone. Exploring the effectiveness of our proposed approaches in other domains would be an interesting avenue for future research.

Secondly, the implementation of our proposed metric relies on the utilization of several NLP tools and resources, including Named Entity Recognition (NER), entity linking, and knowledge graphs. The performance of our metric is therefore influenced by the accuracy and availability of these tools. Further improvements in these areas would enhance the reliability and generalizability of our approach.

## Ethical Considerations

We collected news articles from Chinese official agencies, ensuring that they undergo a thorough double-checking process prior to their release to prevent any inclusion of libelous, racist, or otherwise inappropriate content. While our proposed approach incorporates knowledge graphs and is trained on real news data, we acknowledge the potential for the system to generate fabricated or inaccurate information due to the inherent systematic biases introduced during model pretraining using web corpora. Therefore, we strongly encourage users to exercise caution and critically evaluate the ethical implications of the generated output when applying the system in real-world scenarios.

## Acknowledgement

We thank anonymous reviewers for helpful suggestions. The research work described in this paper has been supported by the National Key Research and Development Program of China under Grant No. 2022ZD0118802

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

## A   Full Contents of Figure 1

The full case is shown in Table 7. We normalize the human-written contents to maximum 1024 tokens, the same number as maximum tokens for generated texts.

## B   Details for Informativeness Human Evaluation

**Texts for human evaluation.**  We carefully examined human-written and model-generated texts in the realm of news articles. To prevent any potential bias caused by language models memorizing specific information, we compiled a dataset consisting of 20 news articles published after January 1, 2023, from reputable sources such as BBC and CNN.

**Model generation setting.**  We adopted the experimental setup outlined in (Xie et al., 2023), where our models were tasked with generating stories based on provided headlines. To encourage greater creativity in the generated texts, we utilized a temperature of 0.8 during the text generation process. Furthermore, we constrained the maximum token count to 1024 to ensure manageable output length.

**Prompt for LLMs.**  We provide the prompts used in our case study(Figure 1) and questionnaires (Table 1). We use "[the news title]" to represent the content of a news title. As different models have different capabilities, we use several prompts.

*GPT-3.5-turbo* : The system prompt for GPT-3.5-turbo is "You are a helpful news writing assistant. You are required to write informative English news based on the given title." and the user prompt is "The title is [the news title]"

*Text-Davinci-003 and GPT3.5-curie* : "You are a helpful news writing assistant. You are required to write an informative English news article based on the given title. Here is the title: [the news title]."

*LLAMA-13B and GPT-3-davinci*:"Title:[the news title]\n News content:"

**Questionnaire preparation and collection.**  The questionnaire comprises a news headline and six news articles, comprising one authored by a human and five generated by the aforementioned models, all derived from the given headline. Since objective quantitative metrics for assessing information content are lacking, we instructed the data annotators to rank the texts according to their perceived level of information instead of assigning quantitative scores. We truncate the news text to up to 1024 tokens, the same number of maximum token numbers of generated texts.

## C   Detailed results of Informativeness Human Evaluation

**Detailed statistic information**   The table 8 contains the statistical information of human-written texts and model-generated texts, which includes the number of words, number of entities, and number of events.

**Win-rate between human and models.**  The figure 4 shows the win rate between humans and models as follows. Since there is a tie-breaker option in the questionnaire, the win rate is likely to be less than 100%.

## D  Case Study

We provide a comprehensive case study comparing the outputs of GPT-3.5-turbo with and without the dynamic entity guide in Table 9. By incorporating the dynamic entity guide, the model generates mentions of entities such as "Falcon 9" and "Starbase Launch Site." This demonstrates the effectiveness of the entity guidance in influencing the generation process and improving the relevancy and informativeness of the generated text.

| | |
|---|---|
| **Input:** Elon Musk sets low expectations before first SpaceX launch of Starship, most powerful rocket ever built | |

**Human:**

Just a few months after NASA introduced the world to the most powerful rocket ever flown to orbit, Elon Musk's SpaceX is prepared to set off its own creation — which could pack nearly twice the power of anything flown before. SpaceX's vehicle, called Starship, is currently sitting on a launch pad at the company's facilities on the southern Texas coastline. The company is targeting liftoff at 8 a.m. CT (9 a.m. ET) on Monday, although it has the ability to take off anytime between 8 a.m. CT (9 a.m. ET) and 9:30 a.m. CT (10:30 a.m. ET). "I guess I'd like to just set expectations low," SpaceX CEO Elon Musk said during a Twitter "Spaces" event for his subscribers Sunday evening. "If we get far enough away from launch pad before something goes wrong, then I think I would consider that to be a success. Just don't blow up the pad." He added: "There's a good chance that it gets postponed since we're going to be pretty careful about this launch." SpaceX has a livestream of the Starship launch here. Folks on the ground near SpaceX's facilities in South Texas can certainly catch an in-person glimpse. Locals are known to line the surrounding beaches in South Padre Island to watch tests, and this launch is sure to draw spectators. SpaceX has repeatedly warned those in the area, however, to stay away from the "Keepout Zone" — the areas directly surrounding the launch site that have been deemed too close to the rocket to be safe during lift off. The "Keepout Zone" includes the coastline south of South Padre Island and stretches a few miles inland. About this mission This will mark SpaceX's first attempt to launch a fully assembled Starship vehicle, building on a years-long testing campaign. Musk has talked about Starship — making elaborate presentations about its design and purpose — for half a decade, and he frequently harps on its potential for carrying cargo and humans to Mars. Musk has even said that his sole purpose for founding SpaceX was to develop a vehicle like Starship that could establish a human settlement on Mars. Additionally, NASA has already awarded SpaceX contracts and options worth several billions of dollars to use Starship to ferry government astronauts to the surface of the moon under the space agency's Artemis program. The inaugural flight test will not complete a full orbit around Earth. If successful, however, it will travel about 150 miles above Earth's surface, well into altitudes deemed to be outer space. Starship consists of two parts: the Super Heavy booster, a gargantuan rocket that houses 33 engines, and the Starship spacecraft, which sits atop the booster during launch and is designed to break away after the booster expends its fuel to finish the mission. The massive Super Heavy rocket booster will give the first blast of power at liftoff. Less than three minutes after takeoff, it's expected to expend its fuel and separate from the Starship spacecraft, leaving the booster to be discarded in the ocean. The Starship will use its own six engines, blazing for more than six minutes, to propel itself to nearly orbital speeds. The vehicle will then complete a partial lap of the planet, reentering the Earth's atmosphere near Hawaii. It's expected to splash down off the coast about an hour and a half after liftoff. Getting here Development of Starship has been based at SpaceX's privately held spaceport about 40 minutes outside Brownsville, Texas, on the US-Mexico border. Testing began years ago with brief "hop tests" of early spacecraft prototypes. The company began with brief flights that lifted a few dozen feet off the ground before evolving to high-altitude flights, most of which resulted in dramatic explosions as the company attempted to land them upright. One suborbital flight test in May 2021, however, ended in success. Since then, SpaceX has also been working to get its Super Heavy booster prepared for flight. The massive, 230-foot-tall (69-meter-tall) cylinder is packed with 33 of the company's Raptor engines. Fully stacked, Starship and Super Heavy stand about 400 feet (120 meters) tall. SpaceX has been waiting more than a year to get FAA approval for this launch attempt. The company, and federal regulators tasked with certifying SpaceX launches won't pose risks to people or property in the area surrounding the launch site, have faced significant pushback from the local community, including from environmental groups. But the Federal Aviation Administration, which licenses commercial rocket launches, announced Friday, April 14, that it granted the company's request for an uncrewed flight test of the rocket out of the SpaceX facilities in South Texas. "After a comprehensive license evaluation process, the FAA determined SpaceX met all safety, environmental, policy, payload, airspace integration and financial responsibility requirements," the agency said in a statement. During a call with reporters last week, an FAA official, who declined to be named for publication, said that the agency has been overseeing SpaceX's compliance with the mitigating actions, some of which are still in the works, even as the company prepares for launch.

**GPT-3.5-turbo**

On April 16th, 2023, Elon Musk, the CEO of SpaceX, set low expectations before the first launch of the Starship, the most powerful rocket ever built. The Starship, designed to carry up to 100 passengers, is intended to transport humans and cargo to the moon, Mars, and beyond. Despite the groundbreaking technology and ambitious goals of the project, Musk has warned that the first launch may not be successful. According to Musk, the first launch will be experimental and the chances of success are uncertain. He acknowledged the high risk associated with space travel and emphasized the importance of taking calculated risks to achieve progress. Musk revealed that the company has conducted numerous tests and simulations to minimize the risks, but there are still many unknown variables that could impact the launch. The Starship project has been in development for several years, and SpaceX has invested billions of dollars into the program. The rocket is designed to be reusable and cost-effective, which could revolutionize the space industry. The success of the project would pave the way for future space exploration and colonization. Despite the potential benefits, Musk has maintained a realistic approach and acknowledged the challenges associated with space travel. The announcement of low expectations may disappoint some supporters, but it shows the company's commitment to safety and responsible innovation. Space exploration is a high-stakes endeavor, and SpaceX's cautious approach reflects their dedication to responsible space travel.

**Text-davinci-003**: On April 16 2023, SpaceX CEO Elon Musk set low expectations for the first launch of the company's Starship, the most powerful rocket ever built. The project has been a work-in-progress for years, and Musk is cognizant of the fact that many of the rocket's components are still untested. In a rare show of modesty, he suggested that the success of the launch was far from certain. "We want to set expectations appropriately," Musk said. "I won't be surprised if it blows up on the launchpad, or it's an epic fail." Musk was equally vocal about the potential of the rocket, stating that if successful, the Starship could one day be used to bring humans to Mars. "We'll probably be able to send people to Mars in five years," said Musk. The Starship is designed to be a fully reusable rocket, making it much more cost-effective than current models. It is powered by three Raptor engines, capable of lifting a payload of more than 100 tons into orbit. The launch is scheduled for early June 2023 and will take place from the company's launch site in Boca Chica, Texas. It will be the first launch of the Starship, and the first time anyone has attempted to launch such a powerful rocket. The launch will be closely watched by the aerospace community, as well as by the general public, as it may be the first step towards a new era of space exploration. It is also a testament to the ingenuity of the SpaceX team and Musk's commitment to pushing the boundaries of space exploration. Regardless of the outcome of the launch, one thing is for certain: Elon Musk has set the bar for space exploration higher than ever before with the Starship.

Table 7: The full texts of human-written news and model generated news mentioned in Figure 1

| model | # of entitiy | # of event | # of word | # of sentence | average rank |
|---|---|---|---|---|---|
| Human | 61.75 | 376.30 | 888.35 | 32.40 | 1.60 |
| Text-Davince-003 | 16.60 | 124.45 | 303.30 | 12.55 | 2.75 |
| GPT-3.5-turbo | 14.05 | 126.90 | 254.60 | 10.35 | 3.25 |
| LLAMA-13B | 40.95 | 159.45 | 460.95 | 21.05 | 4.00 |
| GPT-3.5-curie | 10.80 | 57.55 | 151.60 | 6.55 | 4.40 |
| GPT-3-davinci | 39.45 | 137.70 | 526.40 | 21.20 | 4.60 |

Table 8: Detailed statistic information of human-written texts and model-generated texts

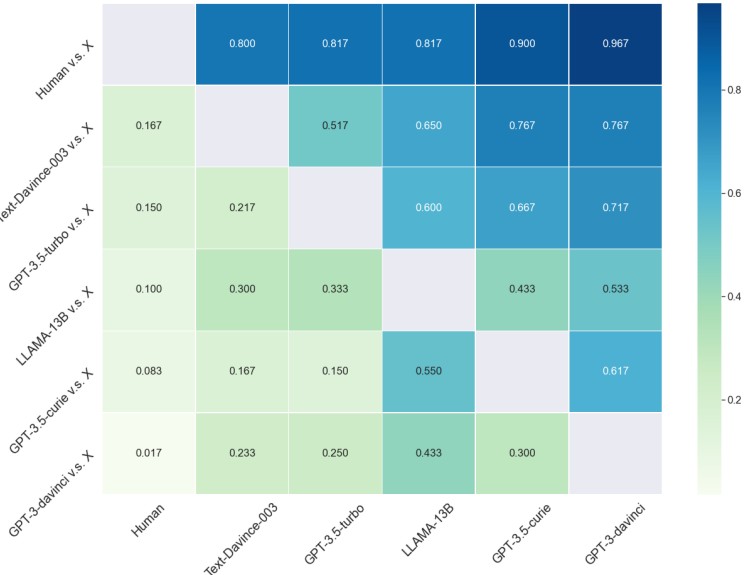

Figure 4: Impact of the number of selected knowledge

| | Input:Elon Musk sets low expectations before first SpaceX launch of Starship, most powerful rocket ever built |
|---|---|

**GPT-3.5-turbo**

On April 16th, 2023, Elon Musk, the CEO of SpaceX, set low expectations before the first launch of the Starship, the most powerful rocket ever built. The Starship, designed to carry up to 100 passengers, is intended to transport humans and cargo to the moon, Mars, and beyond. Despite the groundbreaking technology and ambitious goals of the project, Musk has warned that the first launch may not be successful. According to Musk, the first launch will be experimental and the chances of success are uncertain. He acknowledged the high risk associated with space travel and emphasized the importance of taking calculated risks to achieve progress. Musk revealed that the company has conducted numerous tests and simulations to minimize the risks, but there are still many unknown variables that could impact the launch. The Starship project has been in development for several years, and SpaceX has invested billions of dollars into the program. The rocket is designed to be reusable and cost-effective, which could revolutionize the space industry. The success of the project would pave the way for future space exploration and colonization. Despite the potential benefits, Musk has maintained a realistic approach and acknowledged the challenges associated with space travel. The announcement of low expectations may disappoint some supporters, but it shows the company's commitment to safety and responsible innovation. Space exploration is a high-stakes endeavor, and SpaceX's cautious approach reflects their dedication to responsible space travel.

**GPT-3.5-turbo+guided entitiy**

"NASA" "SpaceX" In a surprising turn of events, Elon Musk, the visionary CEO of SpaceX, has taken the unprecedented step of setting low expectations ahead of the highly anticipated inaugural launch of Starship, the most powerful rocket ever constructed. Acknowledging the complexities of space exploration and the inherent risks involved, Musk addressed both NASA and SpaceX enthusiasts, emphasizing the need for cautious optimism. With NASA's long-standing collaboration with SpaceX, this move showcases Musk's commitment to transparency and safety in the pursuit of ambitious space missions. "SpaceX" "Starship" "Falcon 9" The upcoming launch of Starship marks a significant milestone for SpaceX, as the company aims to demonstrate the capabilities of its revolutionary spacecraft. Designed to carry both crew and cargo, Starship is poised to surpass the achievements of its predecessor, the Falcon 9. With its towering height, sleek design, and immense propulsion power, Starship has captured the imagination of space enthusiasts worldwide. This highly anticipated mission will serve as a crucial test for the rocket's performance, paving the way for future interplanetary exploration and potentially revolutionizing space travel as we know it. "Starship" ""Starbase Launch Site "" "Musk" Elon Musk's bold vision for the future of space exploration has materialized in the form of the Starship, which is being assembled at Starbase Launch Site in Boca Chica, Texas. Musk's decision to construct Starship in Texas reflects his dedication to pushing boundaries and taking on monumental challenges. The expansive manufacturing site has become a hive of activity, with engineers and technicians meticulously working on the intricate components of the spacecraft. Musk's presence and hands-on involvement in the project have inspired the team, creating a sense of shared purpose and determination as they strive to make Starship a reality. "Starship" "Musk" "NASA" The upcoming launch of Starship not only represents a major milestone for SpaceX but also signifies a significant partnership between Elon Musk's company and NASA. The collaboration between SpaceX and the renowned space agency has been instrumental in the development and testing of Starship, with NASA providing valuable expertise and resources. This joint effort highlights the importance of public-private partnerships in advancing space exploration capabilities. As Musk and his team prepare for the historic launch, the support and guidance from NASA serve as a testament to the shared commitment to pushing the boundaries of space travel and realizing the ambitious goals set forth by both SpaceX and the agency.

Table 9: Case of generating texts with GPT-3.5-turbo and GPT-3.5-turbo+guided entity. We highlight guided entity in red and place it before each paragraph.