# OpenReview forum: "Towards Informative Open-ended Text Generation with Dynamic Knowledge Triples"
_EMNLP/2023/Conference — EMNLP 2023 Findings_

### Official Review · Reviewer_GfBH · 2023-07-25

**Soundness:** 4

**Excitement:**

4: Strong: This paper deepens the understanding of some phenomenon or lowers the barriers to an existing research direction.

**Paper Topic And Main Contributions:**

The paper introduces a novel approach that leverages a knowledge graph to guide the text generation process, specifically targeting the issue of over-concentrated information in large language models. The combination of a local knowledge filter, dynamic knowledge selector, and knowledge-enhanced text generator provides a unique contribution to the field of open-ended text generation.

**Questions For The Authors:**

In figure 1, what is the prompt or instructions? Is there any possibility that with better prompts, the generated texts from llm will be more informative?

**Reasons To Accept:**

This paper is well organized and written, the structure is logical, the ideas are presented coherently and clearly.
The method, infogen, is well-motivated, the experimental design is elaborate and the result is solid.


**Reasons To Reject:**

This paper is worth accepting

**Reproducibility:**

4: Could mostly reproduce the results, but there may be some variation because of sample variance or minor variations in their interpretation of the protocol or method.

**Reviewer Confidence:**

4: Quite sure. I tried to check the important points carefully. It's unlikely, though conceivable, that I missed something that should affect my ratings.

---

> ### Author Rebuttal · Authors · 2023-08-29
>
> Thanks for your encouragement.
>
> ### Answers to questions
> We provide the prompts used in our case study(Figure 1) and questionnaires (Table 1). We use "[the news title]" to represent the content of a news title. As different models have different capabilities, we use several prompts.
>
> GPT-3.5-turbo is the API version of ChatGPT and it has a system prompt and user prompt. The system prompt for GPT-3.5-turbo is "You are a helpful news writing assistant. You are required to write informative English news based on the given title." and the user prompt is "The title is [the news title]"
>
> Text-Davinci-003 and GPT3.5-curie are finetuned with instruction, which means that they can follow instruction prompts. The prompt for Text-Davinci-003 and GPT3.5-curie is "You are a helpful news writing assistant. You are required to write an informative English news article based on the given title. Here is the title: [the news title]. "
>
> LLAMA-13B and GPT-3-davinci are just language models and they can only continue writing. The prompt for LLAMA-13B and GPT-3-davinci is "Title:[the news title]\n News content:"
>
> We will add these prompts into the revised version.
>
> The better-designed prompts could lead to more informative texts since our method could be seen as a kind of knowledge-enhanced prompt. But our insight, which is "Knowledge-constrained entities are helpful to generate more informative texts", is more important and could be taken into consideration for prompt design.

---

### Official Review · Reviewer_yWwV · 2023-07-31

**Soundness:** 3

**Excitement:**

3: Ambivalent: It has merits (e.g., it reports state-of-the-art results, the idea is nice), but there are key weaknesses (e.g., it describes incremental work), and it can significantly benefit from another round of revision. However, I won't object to accepting it if my co-reviewers champion it.

**Missing References:**

- Event Transition Planning for Open-ended Text Generation.
  Qintong Li, Piji Li, Wei Bi, Zhaochun Ren, Yuxuan Lai and Lingpeng Kong.
  Findings of the Association for Computational Linguistics: ACL 2022, pp.3412-3426.
  https://aclanthology.org/2022.findings-acl.269.pdf

- ChatGPT is not Enough: Enhancing Large Language Models with Knowledge Graphs for Fact-aware Language Modeling.
  Linyao Yang, Hongyang Chen, Zhao Li, Xiao Ding, Xindong Wu.
  https://arxiv.org/abs/2306.11489

- LMExplainer: a Knowledge-Enhanced Explainer for Language Models.
  Zichen Chen, Ambuj K Singh, Misha Sra.
  https://arxiv.org/abs/2303.16537

**Paper Topic And Main Contributions:**

This paper proposed a variant of guided text generation methods to solve the problem of generating text with excessive use of entities/events contained in an input prompt. The proposed method utilizes a knowledge graph to obtain the triples relevant to an input text, and the obtained triples are encoded to use them as additional inputs of the decoder. To use the reliable triples in the proposed method, the authors introduced the local knowledge filter and dynamic entity selector, both of which are BERT-based modules. In the training of the proposed method, the authors used small LLMs (e.g., BERT-base and BART-base) and fine-tuned them with the dataset of news headline generation. In the evaluation, the authors used three datasets, PENS, CNN and ChinaCNN (ChinaCNN data was created by the authors). In addition to typical generation setting using the proposed encoder-decoder architecture, the authors also evaluated the generation quality by modifying input prompts of GPT-3.5. More precisely, the authors giving the output of the dynamic entity selector as additional inputs to the prompt to investigate whether the entities obtained by the dynamic entity selector has a positive effect on the quality of generation. The experimental results show that the proposed method consistently outperformed the GPT-2/BART-based methods and other guided text generation method with respect to the perplexity, Dist-3, ROUGE-L, BERTScore and newly introduced informativeness-based evaluation metric (KC_entity). Human evaluations also support the superiority of the proposed method over GPT2/BART-based methods.

**Questions For The Authors:**

Question A: The total document sizes of PENS and CNN in Table 2 are not the same as those in the original papers (Ao et al. 2021, Hermann et al. 2015). More concretely, Table 2 of this paper shows 113,218 documents in the PENS dataset, but in Section 3.1 of Ao et al. 2021' paper, the authors described, "The PENS dataset contains 113,762 pieces of new articles ...". In addition, Table 2 of this paper shows 92,465 documents in the CNN dataset, but the dataset contains 92,579 documents (= 90,266 + 1,220 + 1,093) based on the statistics in Table 1 of Hermann et al. 2015's paper. Why are there such differences?

Question B: When does Algorithm 1 terminate? There is no termination condition in the while loop of Algorithm 1 (lines 8 to 14 in Algorithm 1).

Question C: Why did the authors use the Wikipedia dumps archived on October, 4, 2022? Since the date is more recent than the dates in the ChinaCNN dataset, it could cover many of the entities in the dataset. Does that make it a fair experimental setting?

**Reasons To Accept:**

- The proposed methods consistently outperformed the baselines on several evaluation metrics in open-ended news headline generation.

- An attempt to give the dynamic entity selector output as additional inputs to the prompt may be beneficial to some readers.

- The authors reported that the two modules, the local knowledge filter and the dynamic entity selector, are both effective for performance improvement.

**Reasons To Reject:**

- The readability of this paper is low due to crucial typos (e.g., "KG-GPT" in Table 4 -> "CK-GPT") and unclear points (e.g., no termination condition in Algorithm 1).

- Some evaluation metric and evaluation settings are favorable to the proposed method. For example, the proposed evaluation metric "KC_entity" uses the entities obtained from a knowledge graph, but the similar concepts are considered in the local knowledge filter and dynamic entity selector of the proposed model. In addition, the number of the triples used in the experiments were adjusted referring to the target evaluation datasets in Section 5.4. Furthermore, the choice of GPT-3.5 model in Section 4.2 is also unnatural. According to the preliminary results in Table 1, GPT-3.5-davinci is better than GPT-3.5-turbo with regard to both informativeness and #entity, but the authors chose GPT-3.5-turbo in Section 4.2 with no explanation.

- The proposed method strongly depends on the triples obtained from a knowledge graph. This indicates that when no triple cannot be obtained from a knowledge graph, the proposed method does not maintain its generation quality. Since the evaluation task in this paper is news headline generation, where triples can be collected relatively easily from a knowledge graph, it is difficult to know how the performance degrades when no triple is available in the proposed method.

**Reproducibility:**

3: Could reproduce the results with some difficulty. The settings of parameters are underspecified or subjectively determined; the training/evaluation data are not widely available.

**Reviewer Confidence:**

3: Pretty sure, but there's a chance I missed something. Although I have a good feel for this area in general, I did not carefully check the paper's details, e.g., the math, experimental design, or novelty.

**Typos Grammar Style And Presentation Improvements:**

- The authors should right-align the numbers in Table 2 and Table 4 for readability.

- Table 7: the caption said "We highlight guided entity in red and place it before each paragraph." but there is no highlighted entities in this table.

---

> ### Author Rebuttal · Authors · 2023-08-29
>
> Thank you for your valuable questions and comments.
> ### Answers to questions
> #### Answer to  question A
> Thank you very much for your detailed question. The reasons that document sizes of PENS and CNN in Table 2 are not the same as those in the original papers are listed as follows:
>
> For CNN dataset, there are 114 empty samples, which means the sample has empty news article content. For example, there is no news content in "fe9b7daaab79553adaea3e9aa086510bca4b1fb3.story". We remove such samples and get 92465 news articles.
>
> For PENs dataset, there are some noisy samples , which is full of advertisement and web link. We removed the article if web link occupies over 50% content of the text.
>
> Follow your suggestion, we will add these details in Chapter 3 Datasets Construction.
> #### Answer to  question B
> It has two termination conditions. While training PLMs, we use an empty sentence(<bos><eos>) at the end of the news. When the text generator outputs an empty sentence, we terminate the generation process. Besides, we have a maximum number of tokens of generated texts, which is set to 1024. When generated texts touch the limit, we terminate the generation process and discard the incomplete results. While prompting LLMs, termination condition is the maximum number of tokens, which is also set to 1024. We will elaborate the termination condition more clear in the revised version.
> #### Answer to  question C
> Wikidata dumps archived on October 4, 2022, is the latest version when we start to annotate the datasets. Because there lack of Chinese entity link tools and models, we use the Wikidata query API service, which uses the current latest Wikidata dumps. Ideally, news should be linked with the Wikidata dump which has the most recent timestamp.  However, it is too costly, and choosing the current latest version is our balanced choice.
>
> ### Reply to reject reasons
> #### Reply to reject reason1
> We are sorry for these typos and we will check our article and correct them.
> #### Reply to reject reason2
> 1. Our questionnaire result shows that human preference is more correlated with the number of knowledge-constrained entities, rather than the length or number of entities. This motivates us to make use of knowledge-constrained entities to enhance the informativeness of generated texts. We conduct human evaluation and the result indicates that our method indeed improves the informativeness of generated texts.
> 2. Experiments with GPT-3.5-turbo indicate that our method works with LLMs. We chose GPT-3.5-turbo as the baseline mainly because its web version, known as ChatGPT, is popular and convenient to conduct pre-experiments. We will conduct more experiments on different LLMs in the revised version.
> #### Reply to reject reason3
> 1. Knowledge graph is a kind of verified and widely-used structured knowledge base. There exist several general knowledge graphs such as Freebase[1], Wikidata[2] and domain specific such as  AgriKG (agricultural)[3], Acekg(academic)[4], MathGraph(education)[5], SemaTyP(drug discovery)[6]. In knowledge-intensieve text generation, it is rare that entities do not appear in the given input, like news headline or questions.
> 2. Our method will degrade to the baseline when no triple is available. However, when no triples can be obtained, it is questionable whether informativeness is required in such a setting.
>
> ### Reply to Missing References:
> Thank you for providing papers that we may have missed. We will read them and cite them in our paper.
>
> ### Reply toTypos Grammar Style And Presentation Improvements:
> Thank you for your advice and we will correct them.
>
> ### References
> [1] Bollacker, K., Evans, C., Paritosh, P., Sturge, T., & Taylor, J. (2008). Freebase: A collaboratively created graph database for structuring human knowledge, In Proceedings of the 2008 ACM sigmoid international conference on management of data.
>
> [2] Vrandečić, D., & Krötzsch, M. (2014). Wikidata: a free collaborative knowledgebase. Communications of the ACM, 57(10), 78-85.
>
> [3] Chen, Y., Kuang, J., Cheng, D., Zheng, J., Gao, M., & Zhou, A. (2019). Agrikg: An agricultural knowledge graph and its applications, In International conference on database systems for advanced applications. Springer.
>
> [4]Wang, R., Yan, Y., Wang, J., Jia, Y., Zhang, Y., Zhang, W., & Wang, X. (2018). Acekg: A large-scale knowledge graph for academic data mining, In Proceedings of the 27th ACM international conference on information and knowledge management.
>
> [5] Zhao, T., Huang, Y., Yang, S., Luo, Y., Feng, J., Wang, Y., Yuan, H., Pan, K., Li, K., Li, H. Et al. (2019). Mathgraph: A knowledge graph for automatically solving mathematical exercises, In an International conference on database systems for advanced applications. Springer.
>
> [6] Sang, S., Yang, Z., Wang, L., Liu, X., Lin, H., & Wang, J. (2018). Sematyp: A knowledge graph-based literature mining method for drug discovery. BMC bioinformatics, 19(1), 193.

---

### Official Review · Reviewer_GVGV · 2023-08-02

**Soundness:** 3

**Excitement:**

2: Mediocre: This paper makes marginal contributions (vs non-contemporaneous work), so I would rather not see it in the conference.

**Paper Topic And Main Contributions:**

The paper introduces the concept of an "over-concentrated information" issue in open-ended text generation, suggesting that generation models may overly focus on the entities present in the input text, rather than generating a diverse range of entities. To tackle this issue, the authors have designed a comprehensive pipeline. This pipeline includes a local knowledge filter, a dynamic entity selector, and a FiD-based generator. The paper further elaborates on how this pipeline can enhance the performance of Language Learning Models (LLMs) such as ChatGPT.
The paper's contributions are twofold: Firstly, it identifies and addresses the "over-concentrated information" issue by proposing a novel framework. Secondly, it offers three automatically constructed datasets to evaluate the proposed problem.

**Questions For The Authors:**

Table 1 offers significant insight that helps to motivate the paper. However, it would be beneficial if the authors could elaborate on the data used and the analysis performed. Providing more detailed information will help to clarify the basis of the motivation and enhance the overall understanding of the paper.

**Reasons To Accept:**

The proposed proplem ("over-concentrated information" ) is interesting and the paper provides some insight to address such an issue.

**Reasons To Reject:**

1.While the proposed problem is indeed significant in the realm of open-ended text generation, I believe the authors could provide a more comprehensive explanation as to why additional entities should be incorporated into the text generation process. From my perspective, a news article should maintain a strong correlation with its title, rather than diverging into a multitude of topics.
2.The pipeline incorporates several existing techniques and models such as NER, RE, and FiD. I would recommend that the authors provide a rationale for their selection of these specific models over other comparable models in the field.
3.The baselines utilized, namely BART, GPT2, and KG-GPT, are somewhat outdated. I would suggest that the authors compare their pipeline with more recent models in the task, such as the top-performing models in the CNN dataset leaderboard.
4.As I understand it, the pipeline employs an iterative process, wherein a single sentence is generated based on the context and predicted entities. I am curious as to whether this process could potentially compromise efficiency.
5.The paper indicates that experiments were conducted on small PLMs and LLMs. However, there seems to be a lack of comparison in the main results (Table 4) and the case study. I would recommend that the authors address this issue for a more comprehensive analysis.

**Reproducibility:**

3: Could reproduce the results with some difficulty. The settings of parameters are underspecified or subjectively determined; the training/evaluation data are not widely available.

**Reviewer Confidence:**

4: Quite sure. I tried to check the important points carefully. It's unlikely, though conceivable, that I missed something that should affect my ratings.

---

> ### Author Rebuttal · Authors · 2023-08-29
>
> Thank you for your valuable questions and comments.
> ### Answers to questions
> Following your suggestions, we will add additional detailed information and analysis in the revised version. The following table contains the statistical information of human-written texts and model-generated texts, which includes the number of words, number of entities, and number of events.
>
> | model            | # of entitiy | # of event | # of word | # of sentence | average rank |
> | ---------------- | ------------ | ---------- | --------- | ------------- | ------------ |
> | Human            | 61.75        | 376.30     | 888.35    | 32.40         | 1.60         |
> | Text-Davince-003 | 16.60        | 124.45     | 303.30    | 12.55         | 2.75         |
> | GPT-3.5-turbo    | 14.05        | 126.90     | 254.60    | 10.35         | 3.25         |
> | LLAMA-13B        | 40.95        | 159.45     | 460.95    | 21.05         | 4.00         |
> | GPT-3.5-curie    | 10.80        | 57.55      | 151.60    | 6.55          | 4.40         |
> | GPT-3-davinci    | 39.45        | 137.70     | 526.40    | 21.20         | 4.60         |
>
> We will also give the win rate between humans and models as follows. Since there is a tie-breaker option in the questionnaire, the win rate is likely to be less than 100%. If needed, we would offer more detailed results of our questionnaire.Follow your suggestion, we will add these detailed information in the revised versions.
>
> | win-rate                | Human  | Text-Davince-003 | GPT-3.5-turbo | LLAMA-13B | GPT-3.5-curie | GPT-3-davinci |
> | ----------------------- | ------ | ---------------- | ------------- | --------- | ------------- | ------------- |
> | Human v.s. X            | /      | 80.00%           | 81.67%        | 81.67%    | 90.00%        | 96.67%        |
> | Text-Davince-003 v.s. X | 16.67% | /                | 51.67%        | 65.00%    | 76.67%        | 76.67%        |
> | GPT-3.5-turbo v.s. X    | 15.00% | 21.67%           | /             | 60.00%    | 66.67%        | 71.67%        |
> | LLAMA-13B v.s. X        | 10.00% | 30.00%           | 33.33%        | /         | 43.33%        | 53.33%        |
> | GPT-3.5-curie v.s. X    | 8.33%  | 16.67%           | 15.00%        | 55.00%    | /             | 61.67%        |
> | GPT-3-davinci v.s. X    | 1.67%  | 23.33%           | 25.00%        | 43.33%    | 30.00%        | /             |
>
>
>
> ### Reply to reject reasons
> #### Reply to reject reason1:
> In our opinion, informativeness means more details and background information in the paper. When we introduce such information, it's unavoidable to introduce additional entities. That is the reason for incorporating additional entities into informative text generation. As you pointed that the news article should maintain a strong correlation with its title. That is the principle that we design our local knowledge filter, which filters knowledge and entities that are correlated with the title.
>
> #### Reply to reject reason2:
> The reasons for NER and entity link(EL) tools:
>
> For English datasets, we choose Flair and BLINK as NER and EL tools for the following reasons:
>     1. Flair and BLINK are widely used in NLP research and application. Google Scholar citation of Flair is 789 and BLINK is 340. The Github star number of Flair is 12k and BLINK is 1.1k.
>     2. They have a promising performance on English news texts. We compare them with other NER and El tools such as SpaCy, NLTK, and tagme on a small subset of our datasets. They have a better performance.
>     3. BLINK offers model parameters and codes officially, enabling us parallelly process our data on our multiple servers.
>
> For Chinese datasets, we choose SpaCy and Wikidata query as NER and EL tools for the following reasons:
>     1. NER and EL tools such as Flair, NLTK, BLINK, and tagme do not support Chinese texts.
>     2. SpaCy has good performance and is easy to use.
>
> The reasons for relation extraction(RE) tools:
>     The intuition of distant supervision is that any sentence that contains a pair of entities that participate in a known relation in KG is likely to express that relation in some way[1]. This method is simple, effective, and widely used in NLP research[2,3,4].
>
> The reasons for FiD models:
>     FiD converts knowledge to natural texts, which is widely used in knowledge-retrieval LM. This method enables us to parallelly train our model. That is the main reason for choosing FiD models.
>
> Following your suggestions, we will add above reasons in the appendix of the revised version.
>
> #### Reply to reject reason3:
> The motivation of our paper is to solve the "Over-concentrated Information", which leads to a low informativeness of texts generated by pre-trained models. From this perspective, we use pre-trained models as our baselines, including PLMs like GPT2 and BART, and LLMs like GPT-3.5-turbo. Other work in this area mainly focuses on the off-prompt and incoherence problems. In the current version, we do not compare the proposed methods with those methods. Follow your suggestion, we will compare our method  with more recent models in the revised version..
>
> #### Reply to reject reason4:
> Iterative generation is a common generation paradigm in the natural language generation tasks. As you point out, Iterative generation can increase additional costs. Fortunately, the average number of entities per sentence is quite small, especially 1.43 for PENS, 2.31 for CNN, and 1.40 for ChinaNews. Therefore, prediction entities for each sentence do not impose much additional costs. Thanks for your questions, we will add the costs into the revised version.
>
> #### Reply to reject reason5:
> Human evaluation is the golden standard metric for open-ended text generation, especially for LLMs. So we report human evaluation results of experiments on PLMs and LLMs in Table 6. Follow your suggestion, we will add the comparison between small PTMs and LLMs in the main results (Table 4). Meanwhile we will show more the case study.
>
> ### References
> [1]Mintz, Mike, et al. "Distant supervision for relation extraction without labeled data." Proceedings of the Joint Conference of the 47th Annual Meeting of the ACL and the 4th International Joint Conference on Natural Language Processing of the AFNLP. 2009.
>
> [2]Chen, Danqi, et al. "Reading Wikipedia to Answer Open-Domain Questions." Proceedings of the 55th Annual Meeting of the Association for Computational Linguistics (Volume 1: Long Papers). 2017
>
> [3]Ji, Shaoxiong, et al. "A survey on knowledge graphs: Representation, acquisition, and applications." IEEE transactions on neural networks and learning systems 33.2 (2021): 494-514.
>
> [4]Sun, Yu, et al. "Ernie 3.0: Large-scale knowledge enhanced pre-training for language understanding and generation." arXiv preprint arXiv:2107.02137 (2021).

---

### Meta-Review · Area_Chair_g3Qm · 2023-09-15

**Recommendation:** 3

**Metareview:**

The reviewers cite the motivation and clarity of the experimental design as strengths of the paper.

There were some concerns about limitations of this work, e.g., whether the resources/models used in the pipeline approach would be effective in another domain, and whether the evaluation used competitive baselines. Discussion with the authors led two reviewers to slightly increase their scores, though excitement scores remained borderline.

---

### Decision · Program_Chairs · 2023-10-07

**Decision:**

Accept-Findings

**Comment:**

The reviewers cite the motivation and clarity of the experimental design as strengths of the paper.

There were some concerns about limitations of this work, e.g., whether the resources/models used in the pipeline approach would be effective in another domain, and whether the evaluation used competitive baselines. Discussion with the authors led two reviewers to slightly increase their scores, though excitement scores remained borderline.